# The effects of fasting on acute ischemic infarcts in the rat

Anna M. Schneider[1,2]*, Alastair M. Buchan[1], Yvonne Couch[3]

**1** Radcliffe Department of Medicine, Acute Stroke Programme, University of Oxford, Oxford, United Kingdom,
**2** Department of Neurology, University Hospital Zürich and University of Zürich, Zürich, Switzerland,
**3** Nuffield Department of Clinical Neurosciences, University of Oxford, Oxford, United Kingdom

* annamaria.schneider@usz.ch

## Abstract

Inflammation is largely detrimental early in the acute phase of stroke but beneficial at more chronic stages. Fasting has been shown to reduce inflammation acutely. This preliminary study aimed to determine whether post-ischemic fasting improves stroke outcomes through attenuated inflammation. After an endothelin-1 lesion was created in the striatum, Wistar rats were subjected to either regular feeding or water-only fasting for 24 hours. Brain damage and central inflammation were measured histologically, while systemic inflammation was assessed through blood analysis. After 24 hours, fasting was found to reduce infarct volume and BBB breakdown, and lower both circulating and brain neutrophils. These findings suggest that fasting may be a beneficial non-pharmacological additive therapeutic option for cerebral ischemia, potentially by reducing inflammation in the acute stage of the disease.

## Introduction

Inflammation is a critical factor in post-stroke recovery. Emerging evidence suggests that inflammatory cells play complex and multiphasic roles after ischemic stroke, with most cell types exhibiting both beneficial and detrimental effects [1]. Increasing data indicate that early-phase inflammation after ischemic stroke is detrimental, and reducing inflammation during the acute phase is likely neuroprotective [1–3]. However, attempts to translate anti-inflammatory agents into ischemic stroke therapy have largely remained unsuccessful at reducing lesion progression [2].

Most of these interventions have been targeted at specific cytokines, chemokines, or inflammatory processes, even though the inflammatory cascade is multifaceted with significant inherent redundancy [4]. Fasting has been demonstrated as an effective intervention to lower inflammation across different species through various mechanisms, including downregulation of the P13K/Akt/mTOR signalling pathway [5–7]. In rodent models, fasting has been shown to reduce inflammation in stroke [8] and cortical injury [9]. Most studies introduced dietary intervention before the brain injury, underscoring its potential as a preventative measure [10,11]. This study, however, investigates fasting as a treatment, initiating it after a focal cerebral ischemia model in rats, and examines its acute effects on infarct volume, BBB breakdown, and inflammation at 24 hours.

**Data Availability Statement:** All relevant data are within the manuscript and its Supporting Information files.

**Funding:** AMB: 21CVD04. Leducq Foundation. https://www.fondationleducq.org/ The funders had

no role in study design, data collection and analysis, decision to publish, or preparation of the manuscript. YC: ARUK-RF2019B-004. Alzheimer's Research UK. https://www.alzheimers.org.uk/research The funders had no role in study design, data collection and analysis, decision to publish, or preparation of the manuscript.

**Competing interests:** AMB is senior medical science advisor and co-founder of Brainomix, a company that develops electronic ASPECTS (e-ASPECTS). The other authors declare no competing conflict of interest. This does not alter our adherence to PLOS ONE policies on sharing data and materials.

# Materials and methods

## Animals

All experimental procedures were approved by the UK Home Office (1986 Animal Act, Scientific Procedures) and conducted in accordance with the local ethical guidelines of the University of Oxford. Wherever possible, the study adhered to the ARRIVE and IMPROVE guidelines for animal and preclinical stroke work [12,13]. Male Wistar Han rats (250–320 g, obtained from Envigo Research Model Services in Blackthorn, England) were housed in individually ventilated cages under a 12-hour light/12-hour dark cycle with ad libitum access to water. Prior to the intervention, access to food was unrestricted, but during the first 24 hours post-stroke, it was either strictly controlled or restricted. The principal investigator began daily handling and weighing of the animals three days before the surgery.

## Sufficient statistical power

Preliminary data on the effects of fasting after stroke are scarce. Power calculations were based on previous work from our group. A sample size of ten rats per group was considered sufficient to detect biologically significant effects of infarct volume, the primary outcome measure, while minimizing the number of animals required to achieve the research objectives of this study.

## Controls

Appropriate control groups were included in all experiments. Control animals had ad libitum access to food and water and did not receive any pharmacological treatment. To control for the stroke itself, sham animals receiving the respective treatments (n = 10 for fasting, n = 10 for control) were included. Sham surgery followed the same protocol as the endothelin-1 (ET-1) model but with a sterile saline injection instead of ET-1. Furthermore, to control for the surgical intervention itself, naïve animals that did not undergo surgery were included (n = 5 for fasting, n = 5 for control). Throughout the chapter, the different study groups are referred to as "stroke", "sham" and "naïve", respectively. Pre-defined inclusion criteria required data on the infarct volume and BBB breakdown as the primary outcome measures, while pre-defined exclusion criteria were a post-procedural welfare score of 7 or higher (**Table 1**) or death. No animal was excluded from this study. The primary outcome measures were infarct volume and BBB breakdown, while the secondary outcome measures were inflammatory markers in the brain and blood.

## Randomization and blinding

To prevent accidental bias and confounding in the *in vivo* experiments, treatment allocation between fasting and feeding was alternated among the animals. Additionally, tissue samples were randomized after collection (YC) by changing animal identification numbers, and blinding was maintained throughout the experiment whenever possible, until data acquisition was complete.

## Welfare assessment

Post-procedural monitoring included a welfare assessment performed every 30 minutes for the first 3 hours after the surgery and then every 12 hours thereafter, following the scoring system described in **Table 1** and the recommendations of the IMPROVE guidelines [7].

**Table 1. Welfare assessment.**

|  | Scoring |
|---|---|
| **Appearance** |  |
| Normal | 0 |
| Lack of grooming | 1 |
| Coat staring/ocular discharge/nasal discharge | 2 |
| Piloerection/hunched posture | 4 |
| **Food and water intake** |  |
| Maintaining body weight within 5% of baseline weight | 0 |
| Weight loss 5–10% | 1 |
| Weight loss 10–15% | 2 |
| Weight loss > 10–15% | 4 |
| Weight loss > 10–15%, sustained for ≥ 48 hours | 12 |
| **Natural behaviour** |  |
| Normal | 0 |
| Minor change in spontaneous activity | 1 |
| Substantial change in spontaneous activity | 2 |
| Restless or very still | 4 |
| Chewing limb, lameness, loss of body supply to leg | 12 |
| **Cerebral function** |  |
| Normal | 0 |
| Minor/moderate circling and/or hemiparesis | 2 |
| Severe circling and/or hemiparesis/hemiplegia | 6 |
| Fitting or ataxia | 12 |
| **Postoperative complications** |  |
| Difficulty in breathing | 4 |
| Excessive bleeding | 4 |
| Infection | 4 |
| Wound breakdown | 4 |

Welfare assessment ranging from 0 to 56. Scores of less than 4 were considered mild, while scores of 7 and above were regarded as severe impairments. A score of 7 or higher was a pre-defined humane endpoint.

## Endothelin surgery

In this experiment, focal cerebral ischemia was induced using ET-1, chosen based on ethical considerations of employing a stroke model that involves less severe surgery, allowing for subsequent fasting. The ET-1 stroke model is a well-validated and widely used method to induce ischemic stroke [14,15]. The stereotaxic microinjection of ET-1 causes a transient reduction in blood flow lasting approximately 40 minutes, resulting in a focal lesion that can be used to study the effects of focal ischemia in the brain. During surgery, the core body temperature of all animals was maintained at 37.0 ± 0.5°C using a rectal thermometer connected to a feedback-controlled heating pad (Harvard Apparatus, Cambourne, UK). Physical parameters, including body temperature and respiratory rate, were monitored and recorded every 5 minutes throughout the procedure. Respiration was maintained between 50 and 60 breaths per minute by adjusting the isoflurane concentration. Focal brain ischemia was induced by injecting ET-1 into the right striatum, similar to a previously described method [9]. Briefly, the rats were deeply anesthetized with 5% isoflurane in 70% $N_2$ and 30% $O_2$, then maintained at 1–2%

isoflurane throughout the procedure. After weighing, blood glucose and ketone bodies were measured, and the head was shaved and disinfected with a 70% ethanol and 30% chlorhexidine solution. The animal was secured in a stereotaxic frame, and a midline incision at the top of the head was made. The needle was then guided to the coordinates of the MCA territory in the right hemisphere's striatum (AP +1.0, ML -3.0, SI -4.0 mm). A small hole was drilled in the skull, and 1 µl of ET-1 (25 pmol) or saline (for the control group) was slowly injected over 2 minutes. The head wounds were cleaned and closed (4–0 Vicryl Rapide Undyed 1x18" P-3, Somerville, US). Animals received 0.05 mg/kg marcaine in the wound to alleviate pain and 2 mL of saline solution subcutaneously. The rectal thermometer was removed, isoflurane was turned off, and the gases were switched to 0% $N_2O$ and 100% $O_2$. Upon awakening, the animals were placed in a pre-warmed cage on a heating mat. They were closely monitored for 3 hours after awakening, with post-surgical welfare checks every 30 minutes.

## Treatment

Animals in the fasting group had no access to food for 24 hours from the point of suture closure, with ad libitum access to water (n = 11). The control group had free access to food and water (n = 10).

## Tissue processing

After 24 hours from the start of treatment, rats were deeply anesthetized with 5% isoflurane in 70% $N_2$ and 30% $O_2$, and their weight was measured. Their blood glucose and β-hydroxybutyrate ketone levels were also measured. This was done using blood from a tail prick (minimum 1.2 µl) that was immediately applied to a test strip and inserted into the glucose- and ketone-monitoring meter, providing results within 10 seconds (On Call GK Dual Blood Glucose & Ketone Monitoring System, Acon Laboratories, San Diego, CA, USA). The rats were then killed by intraperitoneal injection of pentobarbital (800 mg/kg). Five mL of blood was drawn from the heart into EDTA tubes, and a full blood count was performed the same day (Laboratory Haematology, John Radcliffe Hospital, Oxford OX3 9DU, UK). The animals were transcardially perfused with heparinized saline and 4% PFA in PBS. Brains were collected, post-fixed in 4% PFA in PBS overnight, and then transferred to a 30% sucrose solution in PBS. For the naïve group, fresh non-perfused brains were collected and sliced into 2 mm thick coronal sections using an ice-cold stainless-steel matrix (Kent Scientific). Tissue samples (1 mm$^2$) from the striatum and cortex of both hemispheres were snap-frozen on dry ice.

## Immunohistochemistry

Perfused brains were dehydrated through graded ethanol solutions and embedded in OCT mounting medium before being snap-frozen on dry ice. Using a cryostat, 10 µm-thick serial sections in the coronal plane were collected from 0.5 mm anterior to posterior of the lesion on gelatinized slides and stored at -80˚C until further use. Brain sections were rehydrated through graded ethanol solutions. Nonspecific binding was blocked using 10% serum from the species in which the secondary antibody was raised (diluted in PBS) for 1 hour at room temperature (RT) and incubated with the primary antibody in PBS at 4˚C overnight. Afterwards, sections were rinsed in PBS, and a secondary antibody was added at 1:500 in PBS for 45 minutes at RT. Antibody binding was visualized with 3,3'-diaminobenzidine. Sections were dehydrated through graded ethanol solutions and cleared with Histo-Clear II (National Diagnostics, HS2021GLL). The slides were mounted with glass coverslips using an anti-fade fluorescence mounting medium (Dako, USA) and imaged with a microscope scanner (Manual Whole Slide Imager 2017b-31, Olympus Life Science) at 10x magnification.

## Infarct volume

Cresyl violet was used to stain five brain sections per animal, which were +1.28 mm, +0.6 mm, +0.12 mm, -0.12, and -0.48 mm away from the bregma, respectively. To calculate the lesion volume, infarct areas were multiplied by the distance between the brain sections. Lesion volume was then presented as a percentage of the contralateral hemisphere.

## Blood analysis

For full blood count analysis, 5 mL of whole blood was drawn directly out of the heart into EDTA tubes and analyzed the same day (Laboratory Haematology, John Radcliffe Hospital, Oxford, OX3 9DU, UK).

## Statistical analysis

Statistical analysis was performed using Prism 6 (Graphpad, USA). For analysis of infarct volume, BBB breakdown, and neutrophil infiltration into the brain, an unpaired t-test was applied. For weight, blood glucose and ketone bodies, microglia and astrocyte counts, and full blood cell count, a 2-way ANOVA with Dunnett's multiple comparisons test was used. The D'Agostino and Pearson normality tests were performed on all data, and the appropriate statistical tests were chosen based on the normality of the data. An $\alpha$ level of 0.05 was considered statistically significant, and the results are presented as mean ± standard deviation (SD). * $p < 0.05$, ** $p < 0.01$, *** $p < 0.001$, **** $p < 0.0001$.

# Results

## Fasting reduces infarct volume

To investigate and compare the effects of fasting on stroke volume, lesions were stained with cresyl violet 24 hours post-stroke. Stroke volume for each brain was calculated by determining the infarcted areas, multiplying them by the corresponding distance from bregma, and presenting the result as a percentage of the volume of the ipsilateral hemisphere. The relative percentages of all brain sections from the fasting group were pooled and compared with the control group. There was a statistically significant difference in infarct volume between the groups (t-test; p = 0.0071, fasting: 2.685 ± 2.574; control: 7.114 ± 3.921) (**Fig 1A and 1B**).

## Fasting reduces BBB breakdown

Next, the effect of fasting on BBB integrity was investigated. At 24 hours, brains were processed for IHC, and BBB compromise was assessed using IgG staining to visualize serum proteins in the CNS. Ten animals per group were studied, with three sections per brain analyzed at 1.28 mm, 0.12 mm, and -0.48 mm from bregma, respectively. IgG-positive staining was identified, and the IgG-positive area was calculated as a percentage of the contralateral hemisphere. The mean level of BBB breakdown was significantly lower in fasted animals compared to the control group (t-test, p = 0.0016; fasting: 4.859 ± 1.590; control: 9.523 ± 3.648) (**Fig 2A and 2D**).

## Fasting does not affect CNS resident immune cells

Microglia and astrocytes (identified as Iba-1 and GFAP-positive cells, respectively) were quantified at 24 hours within a 1 mm$^2$ area of the cortex and striatum. For microglial counts in the striatum, there was no main effect of stroke (2-way ANOVA; p = 0.0525), no main effect of treatment (p = 0.6224), and no interaction between the effects (F(1,37) = 1.976, p = 0.1618). Šidák's multiple comparisons test revealed a significant increase in microglial numbers in

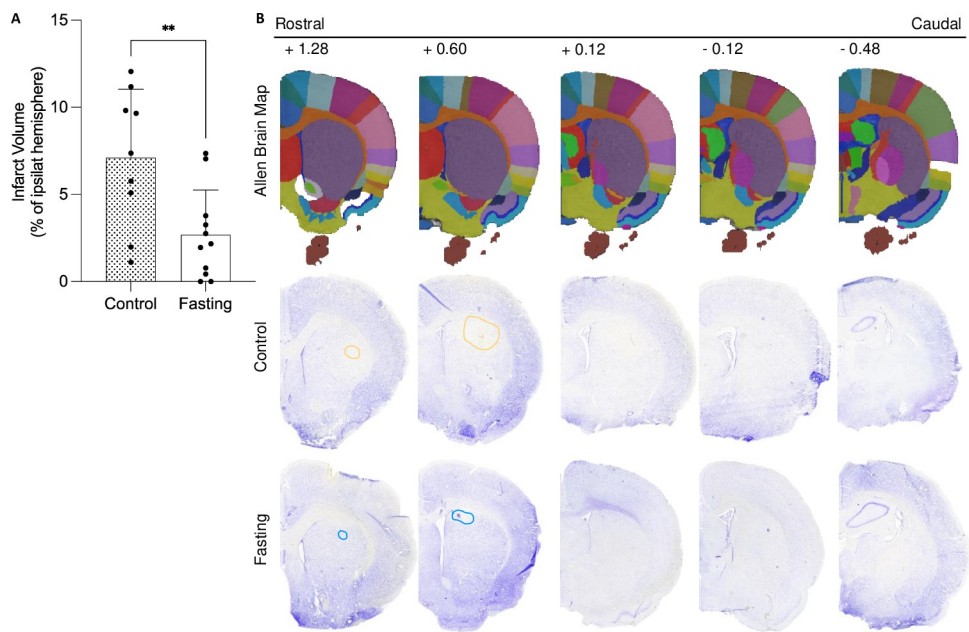

**Fig 1. Fasting reduces infarct volume. (A)** Comparison of infarct volume between control and fasted animals. **(B)** Representative images of both control and fasted animals. Data are presented as mean ± SD. **p<0.001. Sample sizes: n = 9 for the control and n = 11 for the fasted group. The scale bar represents 1 mm.

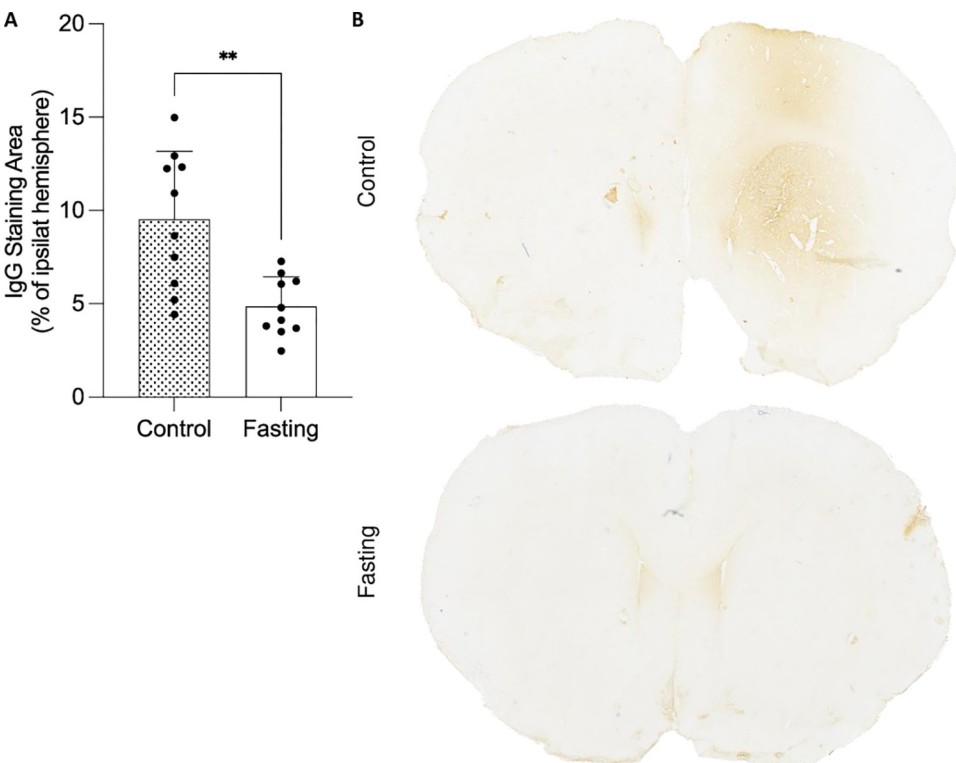

**Fig 2. Fasting reduces BBB breakdown. (A)** Visualization of BBB breakdown through IgG-positive staining. **(B)** Representative images of control and fasted animals. Data are presented as mean ± SD. **p<0.001. Both groups had a sample size of n = 10. The scale bar represents 1 mm.

control animals subjected to stroke compared to the sham group (**Fig 3B**). In the cortex, neither stroke nor treatment had a significant effect on microglial numbers (**Fig 3A**). Similarly, neither stroke nor treatment affected astrocyte numbers in the cortex or striatum (**Fig 3C and 3D**).

### Fasting reduces neutrophil infiltration into the striatum

The mean number of neutrophils in the CNS was calculated, showing a statistically significant difference between the two treatment groups (t-test; p = 0.0249; fasting: 50.91 ± 53.11; control: 145.7 ± 116.7) (**Fig 4A**).

### Fasting reduces circulating neutrophils

At 24 hours post-stroke, a full blood count was performed to assess neutrophil numbers, white blood cell (WBC) counts, lymphocyte counts, and the neutrophil/lymphocyte (N/L) ratio. There was no significant effect of surgery on neutrophils (2-way ANOVA; p = 0.5823), no effect of treatment (p = 0.0889), and no interaction between these factors (p = 0.2611). However, Šidák's multiple comparisons test showed that fasting significantly reduced circulating neutrophil number in stroke animals (p = 0.0400) (**Fig 4D**). For WBC counts, lymphocyte numbers, and the N/L ratio, there was a significant main effect of the surgical intervention, but no effect of treatment and no interaction between the two factors (**Fig 4C, 4E and 4F**).

### Fasting decreases body weight

Bodyweight measurements were recorded immediately before the surgical intervention and again at 24 hours, marking the end of the experiment. The aim was to determine whether fasting affects body weight in animals that have experienced a stroke. There was no main effect of surgery (2-way ANOVA; p = 0.3582), but there was a main effect of treatment (p<0.0001) and an interaction between the two factors (F(2, 45) = 4.130, p = 0.0225). Šidák's multiple comparisons test revealed that fasting significantly reduced body weight in naïve, sham, and stroke animals (p<0.0001 for all) (**Fig 5A**).

### The effects of fasting on blood glucose levels and ketone bodies

The next objective was to investigate whether fasting affects blood glucose and ketone levels in animals that have experienced a stroke. At 24 hours post-stroke, blood glucose and ketone body levels were measured. There was a significant main effect of surgery (2-way ANOVA; p = 0.0412) and treatment (p<0.0001), but no significant interaction between the two (p = 0.0661). Šidák's multiple comparisons test showed significant decreases in glucose levels in fasted animals across all intervention groups (p = 0.0115 for naïve, and p<0.0001 for sham and stroke animals) (**Fig 5B**). For ketone body levels, there was a trend toward a main effect of surgery (p = 0.0631), a main significant main effect of treatment (p<0.0001), and no interaction between the two (F(2,45) = 2.262, p = 0.1158). Šidák's multiple comparisons test revealed that fasting significantly increased ketone bodies in all treatment groups (p = 0.0105 for naïve, and p<0.0001 for sham and stroke animals) (**Fig 5C**).

### Discussion

Unlike pharmacological treatments, fasting may offer benefits as it lacks a significant side-effect profile [5,16]. However, the inherent biochemical and metabolic differences between rodents and humans must be taken into account when expanding and translating these studies [17]. The dietary habits of stroke patients are influenced by clinical outcomes. For instance, dysphagia is a common issue, but no controlled studies have been conducted to assess its

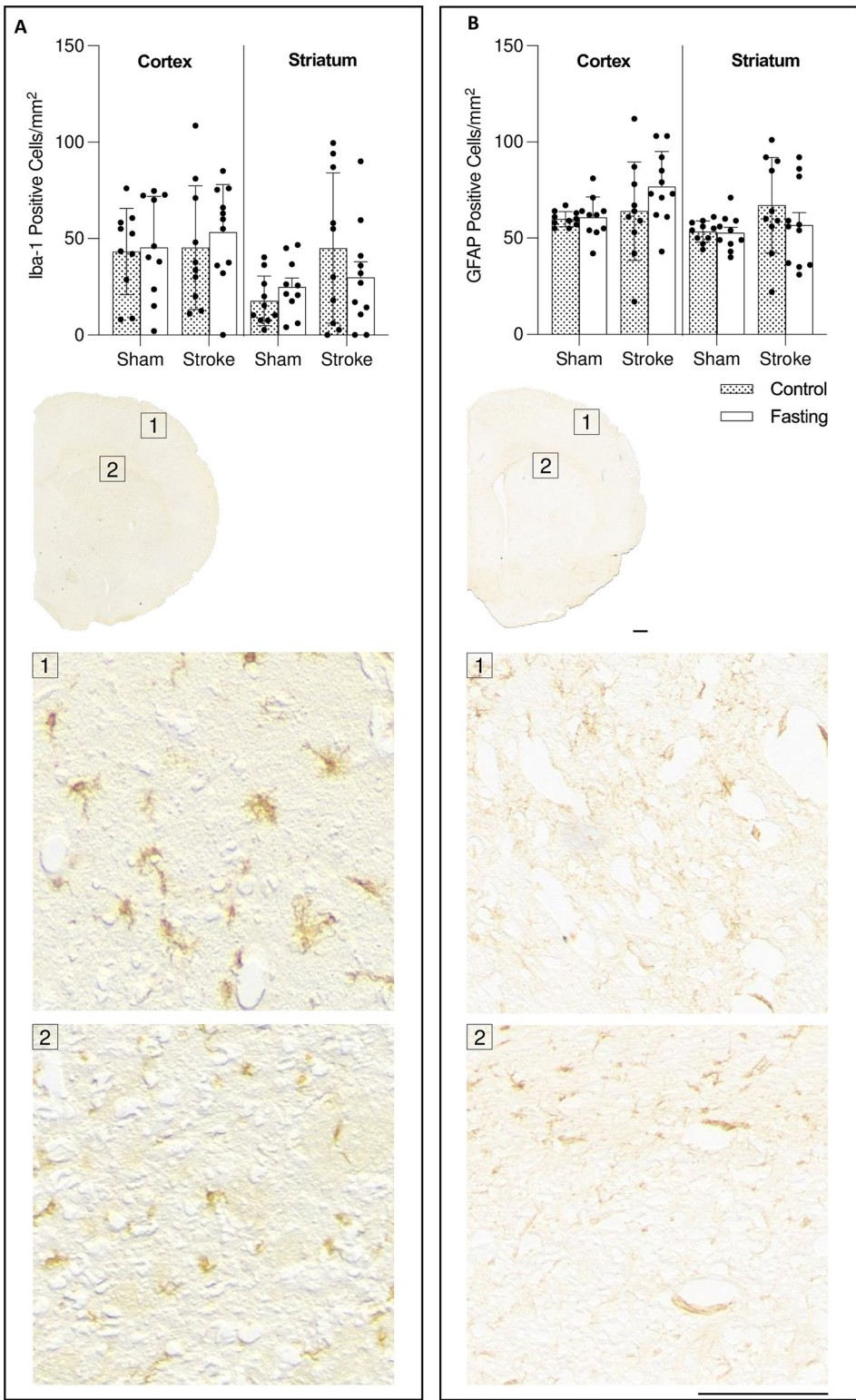

**Fig 3. Fasting does not affect resident immune cells.** Microglial numbers in the **(A)** cortex and **(B)** striatum, and astrocyte numbers in the **(C)** cortex and **(D)** striatum. **(E, F)** Representative images showing microglial or astrocyte-staining, respectively, in the cortex and corpus callosum of a fasted rat. Data are presented as mean ± SD. *p<0.05. Sample size: n = 10/group for all groups. The scale bar represents 1 mm.

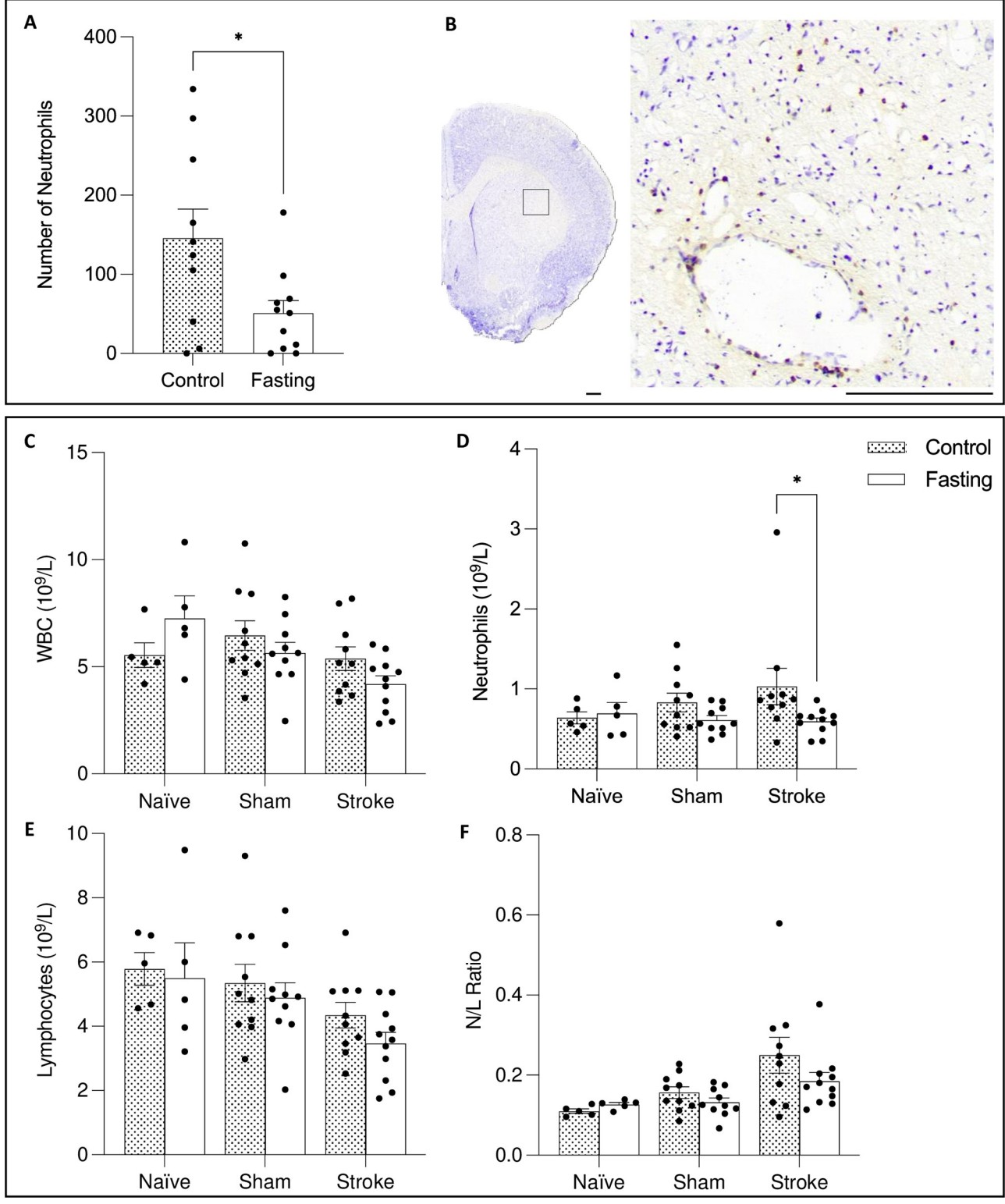

**Fig 4. Fasting reduces central neutrophil infiltration and reduces blood neutrophil counts.** (A) Neutrophils infiltrating the striatum in post-ischemic rats. (B) Representative image of infiltrating neutrophils. (C) WBC count, (D) neutrophils, (E) leukocytes, and (F) N/L ratio. Data are presented as mean ± SD. For histological analysis: n = 10/group for all groups. For blood analysis, n = 5 for naïve groups, n = 10 for sham and stroke groups. *p<0.05. The scale bar represents 1 mm.

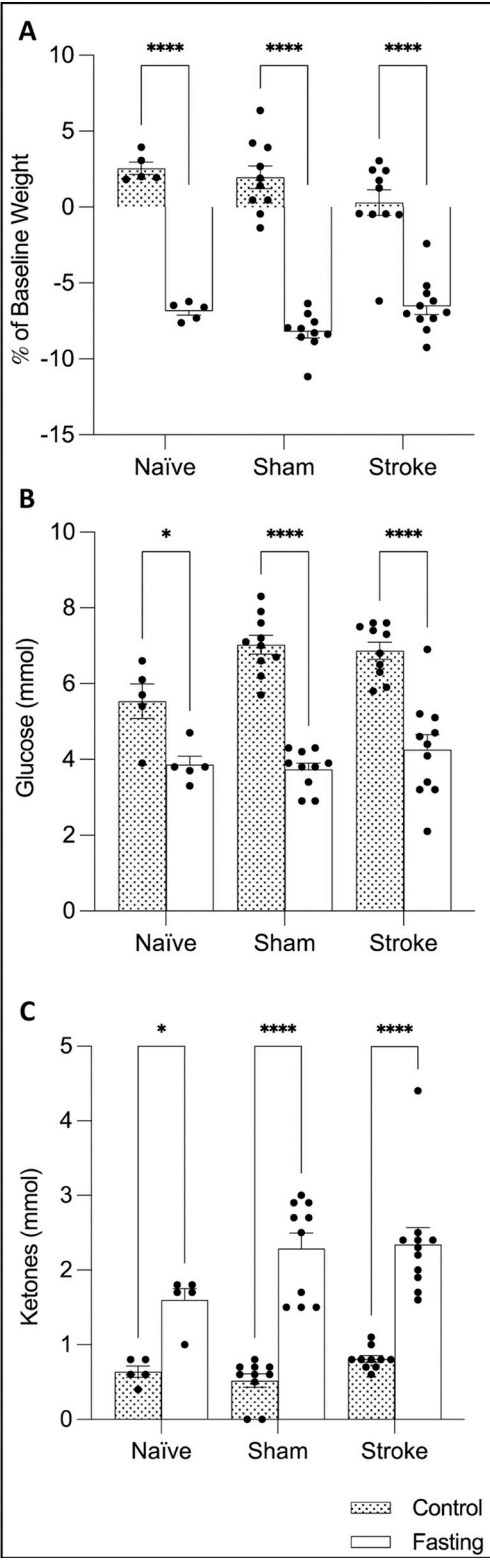

**Fig 5. Fasting reduces body weight and blood glucose levels, while increasing ketone bodies at 24 hours. (A)** Body weight, **(B)** blood glucose, and **(C)** ketone bodies. Results are presented mean ± SD. *p<0.05, ****p<0.0001. Sample sizes: n = 5 in naïve groups, n = 10 in sham and stroke groups.

effects on stroke outcomes [18,19]. In contrast, in rodent studies, researchers often ensure the animals are eating properly as a sign of good recovery, and animals that fail to eat due to severe injury are frequently excluded from the studies. This study aimed to investigate the impact of a 24-hour water-only fast in rats following ET-1-induced cerebral ischemia. Results showed that fasting significantly reduced infarct volume and BBB breakdown and offset the increased circulation and infiltration of neutrophils into the striatum caused by ischemia. These findings suggest that fasting could be a promising approach to mitigating stroke injury by potentially reducing inflammation and ameliorating BBB breakdown.

Inflammation is a critical component of the post-stroke environment, and fasting is recognized for its anti-inflammatory properties [1–3]. Weston and colleagues explored the temporal relationship between ischemic damage and neutrophil numbers in the ET-1 stroke model [20]. They found that infarct volume and neutrophil infiltration into the brain peaked at three days and that neutrophil numbers positively correlated with the infarcted tissue [20]. In our study, we observed that at 24 hours, alongside increased neutrophils in the blood, neutrophils had infiltrated the brain, and this infiltration was reduced by the fasting period following the stroke. Future studies with a 72-hours endpoint could determine if the effect persists beyond the acute phase.

Evidence from cardiac research suggests that fasting could be beneficial [21], but comparisons to more sustainable, longer-term options, such as caloric restriction or timed feeding (intermittent fasting), would also be valuable. Given the dynamic nature of the post-stroke inflammatory response, it is possible that varying the feeding strategy in the immediate post-stroke period could help maintain a reduced inflammatory response—starting with fasting, followed by timed feeding, and then caloric restriction. However, these investigations were beyond the scope of this preliminary study.

The current study did not find any effect of fasting on the numbers of microglia and astrocytes at 24 hours. Microglia typically react to hypoxic stimuli with a brief, localized increase in number and change in morphology, while astrocytes exhibit more long-term, widespread astrogliosis [22,23]. The study's approach of quantifying, but not morphologically defining microglia, likely limits our ability to detect subtle, subacute changes in CNS inflammation.

To advance this preliminary study and broaden our understanding of using fasting as a treatment option for ischemic stroke, we recommend a three-fold approach. First, larger preclinical studies should be conducted to determine the optimal fasting regimen and duration [24]. These studies will also help establish more effective treatment readouts–such as bloodborne biomarkers or novel imaging techniques–that can be easily translated across species. The second step is to translate these findings from rodents to humans. Small clinical trials will help determine treatment efficacy in a heterogeneous patient population and assess feasibility in acute stroke care settings. Finally, the new intervention should be compared to the current gold-standard treatment to evaluate its true clinical benefit.

In conclusion, this study examined the effects of fasting on infarct volume, BBB integrity, and post-ischemic inflammation at 24 hours. The results are promising and suggest that fasting could be a potential adjunct treatment for ischemic stroke. Whilst more research is needed to explore the effects of fasting in the chronic phase post-injury, this initial dataset indicates that fasting could be a valuable, low-cost, and widely accessible treatment option without the risk of pharmacological side effects.

## Supporting information

**S1 Checklist.**
(PDF)

## Acknowledgments

The authors wish to express their gratitude to Liam Silvera at the University of Caltech for his valuable assistance.

## Author Contributions

**Conceptualization:** Anna M. Schneider, Alastair M. Buchan, Yvonne Couch.

**Data curation:** Anna M. Schneider.

**Formal analysis:** Anna M. Schneider.

**Methodology:** Anna M. Schneider.

**Resources:** Alastair M. Buchan.

**Supervision:** Alastair M. Buchan, Yvonne Couch.

**Validation:** Anna M. Schneider, Alastair M. Buchan, Yvonne Couch.

**Visualization:** Anna M. Schneider.

**Writing – original draft:** Anna M. Schneider.

**Writing – review & editing:** Alastair M. Buchan, Yvonne Couch.

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
