## [Decision Letter · Decision Letter 0]

28 Jun 2023

PONE-D-23-12918The effects of fasting on ischemic infarcts in the ratPLOS ONE

Dear Dr. Schneider,

Thank you for submitting your manuscript to PLOS ONE. After careful consideration, we feel that it has merit but does not fully meet PLOS ONE’s publication criteria as it currently stands. Therefore, we invite you to submit a revised version of the manuscript that addresses the points raised during the review process.

Based on the reviewers' suggestions, the paper needs major revision.  The reviewers' comments can be found below.

We look forward to receiving your revised manuscript.

Kind regards,

Tanja Grubić Kezele, Ph.D., M.D.

Academic Editor

PLOS ONE

“AMB is senior medical science advisor and co-founder of Brainomix, a company that develops electronic ASPECTS (e-ASPECTS). The other authors declare no competing conflict of interest.”

Reviewers' comments:

Reviewer's Responses to Questions

**Comments to the Author**

1. Is the manuscript technically sound, and do the data support the conclusions?

Reviewer #1: No

Reviewer #2: Partly

2. Has the statistical analysis been performed appropriately and rigorously? 

Reviewer #1: No

Reviewer #2: Yes

3. Have the authors made all data underlying the findings in their manuscript fully available?

Reviewer #1: Yes

Reviewer #2: Yes

4. Is the manuscript presented in an intelligible fashion and written in standard English?

Reviewer #1: Yes

Reviewer #2: Yes

5. Review Comments to the Author

Reviewer #1: Estimating infarct volume at 1 day is not sufficient in ischemia studies. They need to conduct longer term studies to confirm that effects of fasting are long-lasting. This important as mice eat less food during the first day after ischemia.

The scope of this study is very skimpy. Outcomes are disappointingly low in number. They need to show if fasting leads to neurological benefits by studying at least motor function after ischemia.

The rigor of the data is low. In Fig. 1, infarct volume in control group was spread from 1 % to 12% in various animals. But, the SEM was shown as only ~1%.

Rigor of immunostaining is poor. The Iba-1 stained and GFAP stained images are not acceptable for counting the cell number. I can’t see any cells in the sections they provided. The

For Fig. 1, it is not sufficient to give one CV stained section for each group. Provide serial brain sections for at least one animal of each group.

Need to present infarct volume as mm3 rather than % of ipsilateral hemisphere.

It is a better practice to provide SD than SEM.

Reviewer #2: This study explores the intriguing and significant impact of a 24-hour fasting regimen on inflammation and brain injury induced by stroke. The research has been conducted with meticulous attention; however, this reviewer would like to offer a few suggestions for further enhancing the study prior to publication.

Critique:

1. Incorporating dietary interventions before the occurrence of brain injury holds translational relevance, as it has been demonstrated to prime the organism for protection against major stress conditions. While post-treatment (fasting) serves as an intervention, pre-treatment acts as a preventive measure. Both approaches hold translational significance.

2. The Results section requires improvement. I'm uncertain why the figure legends are placed within the Results section. Additionally, the authors should provide detailed explanations of the results instead of presenting them as brief, one-line statements.

3. There are several significant previous studies in this field that have not been cited. I strongly recommend that the authors discuss these studies in both the introduction and the discussion sections, rather than solely focusing on a few particular studies. This will strengthen the overall narrative and emphasize the relevance of the current study in relation to other existing research.

4. The immunohistochemistry images appear unclear. It is necessary for the authors to provide clear and distinct images that clearly indicate the different cell types quantified in Figure 3. Additionally, it is advisable to include these images from all animals in the supplementary section for comprehensive visualization.

6. PLOS authors have the option to publish the peer review history of their article (what does this mean?). If published, this will include your full peer review and any attached files.

Reviewer #1: No

Reviewer #2: No

---

## [Author Response · Author response to Decision Letter 0]

18 Nov 2023

Reviewer #1: 

1. Estimating infarct volume at 1 day is not sufficient in ischemia studies. They need to conduct longer term studies to confirm that effects of fasting are long-lasting. This important as mice eat less food during the first day after ischemia.

We thank the reviewer for taking the time to carefully review our manuscript. Whilst we agree that it is important, especially in the context of the long-term consequences of stroke, to determine the effects of any interventions on chronic outcomes of ischemia we would also like to highlight that was not the main aim of this study. This preliminary investigation aimed only to determine the effects of fasting on the acute phases post-ischemia to determine whether a longer study would prove beneficial. We would also be keen for the reviewer to note that the model used is the endothelin model in rats, not mice. This model has been used because it produces a particularly robust, reproducible and – most importantly – mild injury. The aim was to study a defined infarct area in the first instance. This model produces limited behavioral deficits and no obvious hypophagia in the first 24 hours (1, 2). Ongoing studies in our laboratory aim to look at the chronic effects of acute fasting and caloric restriction on the outcomes of ischemia but these were not the scope of the current study. We feel that it remains important for scientific collaboration, reproducibility and progress that data are published when they form a complete story. This study aimed to investigate the acute effects of fasting on the acute stages of injury. In line with this, we have adjusted the title to include the word ‘acute’ to make this clear. 

2. The scope of this study is very skimpy. Outcomes are disappointingly low in number. They need to show if fasting leads to neurological benefits by studying at least motor function after ischemia.

As mentioned above, this study aimed to investigate the acute effects of fasting on the molecular biology of the infarct. We have highlighted this in our adjusted title. The ongoing work in our laboratory aims to investigate the role of fasting on more detailed behavioural outcomes such as paw reaching, motor learning and neurological scoring. We also aim to use other models of stroke, including the middle cerebral artery occlusion model, which produces more significant behavioral deficits than the endothelin model and as such would be more amenable to this kind of testing. As we hope the reviewer appreciates, many projects are produced with limited time and funds and must be published to secure further funding to be expanded. This is a preliminary data set that we feel would benefit the field of stroke research. 

3. The rigor of the data is low. In Fig. 1, infarct volume in control group was spread from 1 % to 12% in various animals. But, the SEM was shown as only ~1%.

As noted by previous studies from our group as well as others, within-group variability of stroke volume has been discussed extensively in the literature. The reasons for this include the collateral system of the animal studied and the energy availability at the time point of study, amongst others. Since the SEM quantifies uncertainty in estimates of the mean, the SD indicates the actual dispersion of the data from the mean. Accordingly, one can still have a relatively high variability in the data with a low SEM. We have, based on comment number 7 and in accordance with the current statistical reporting guidelines, amended the reporting from using SEM to SD.

4. Rigor of immunostaining is poor. The Iba-1 stained and GFAP stained images are not acceptable for counting the cell number. I can’t see any cells in the sections they provided.

5. For Fig. 1, it is not sufficient to give one CV-stained section for each group. Provide serial brain sections for at least one animal of each group.

We appreciate the reviewer’s comments. Whilst many groups do use serially stained sections using TTC, many use cresyl violet, toluidine blue or haematoxylin and provide a single image as an example of maximum infarct volume, this is standard within the field of pre-clinical stroke research. Some examples are provided in the table below. It should be noted that within this list, researchers such as Kate Lambertsen are prominent within the field of stroke research and regularly represent their lesions in this way.

Sudhir Karthikeyan et al 2018: Characterizing Spontaneous Motor Recovery Following Cortical and Subcortical Stroke in the Rat.

Bettina Hjelm Clausen et al 2017: Fumarate decreases edema volume and improves functional outcome after experimental stroke.

Bettina Hjelm Clausen et al 2016: Conditional ablation of myeloid TNF increases lesion volume after experimental stroke in mice, possibly via altered ERK1/2 signaling.

Hima C. S. Abeysinghe et al 2014: Brain Remodelling following Endothelin-1 Induced Stroke in Conscious Rats.

Hiroshi Kamada et al 2007: Bad as a Converging Signaling Molecule between Survival PI3-K/Akt and Death JNK in Neurons after Transient Focal Cerebral Ischemia in Rats.

Kate Lykke Lambertsen et al 2004: A Role for Interferon-Gamma in Focal Cerebral Ischemia in Mice.

Kate Lykke Lambertsen et al 2004: A Quantitative Study of Microglial—Macrophage Synthesis of Tumor Necrosis Factor during Acute and Late Focal Cerebral Ischemia in Mice.

Kate Lykke Lambertsen et al 2002: Microglial—Macrophage Synthesis of Tumor Necrosis Factor after Focal Cerebral Ischemia in Mice is Strain Dependent.

6. Need to present infarct volume as mm3 rather than % of ipsilateral hemisphere.

In our previous studies, we have used MRI as a means to analyse infarct volume. As part of these studies, and in studies by others it has been noted that rat brain volume varies by body weight and as such, calculating lesion volume as a percentage of the rats' own brain proved to be an effective way to counter this variability and controls for inter-individual differences in brain volumes (3). This also allows analysis to take into account edema and was originally published by Swanson in 1990 (4).

7. It is a better practice to provide SD than SEM.

We thank the reviewer for this comment and adjusted the reporting of the statistical analysis from SEM to SD accordingly. Please also see our response to comment 3 for further justification.

Reviewer #2: 

This study explores the intriguing and significant impact of a 24-hour fasting regimen on inflammation and brain injury induced by stroke. The research has been conducted with meticulous attention; however, this reviewer would like to offer a few suggestions for further enhancing the study prior to publication. Critique:

1. Incorporating dietary interventions before the occurrence of brain injury holds translational relevance, as it has been demonstrated to prime the organism for protection against major stress conditions. While post-treatment (fasting) serves as an intervention, pre-treatment acts as a preventive measure. Both approaches hold translational significance.

We thank the reviewer for their stimulating comment. We agree that there is some degree of translational relevance in that fasting as a matter of habit may improve recovery if the individual were to have a stroke, our main goal is to help those who have already had a stroke, irrespective of their prior dietary habits. We have highlighted this in the text and adjusted to make this clear, and pointed out that pre-stroke fasting may be a beneficial preventative measure: “Most studies introduced the dietary intervention before the brain injury, which highlights its potential as a preventative measure. This study, however, introduces fasting as a treatment by introducing it after a focal model of cerebral ischemia in rats and investigates its acute effects on infarct volume, BBB breakdown, and inflammation at 24 hours.“

2. The Results section requires improvement. I'm uncertain why the figure legends are placed within the Results section. Additionally, the authors should provide detailed explanations of the results instead of presenting them as brief, one-line statements.

PLOS One Journal asks the author to provide the Figure in the text of the manuscript, immediately following the paragraph in which the figure is first cited. Considering the comment on providing more detailed descriptions of the results, we extended the explanation of the results accordingly.

3. There are several significant previous studies in this field that have not been cited. I strongly recommend that the authors discuss these studies in both the introduction and the discussion sections, rather than solely focusing on a few particular studies. This will strengthen the overall narrative and emphasize the relevance of the current study in relation to other existing research.

We appreciate the reviewer’s comment and have expanded the references with relevant literature on fasting and ischemic stroke both, in the introduction and the discussion section. 

4. The immunohistochemistry images appear unclear. It is necessary for the authors to provide clear and distinct images that clearly indicate the different cell types quantified in Figure 3. Additionally, it is advisable to include these images from all animals in the supplementary section for comprehensive visualization

We thank the reviewer for their comments, as both reviewers have highlighted the quality of the images provided we have updated these and provided higher-powered images for the revised manuscript.

1. Abeysinghe HCS, Roulston CL. A Complete Guide to Using the Endothelin-1 Model of Stroke in Conscious Rats for Acute and Long-Term Recovery Studies. Methods Mol Biol. 2018;1717:115-33.

2. Weston RM, Jones NM, Jarrott B, Callaway JK. Inflammatory cell infiltration after endothelin-1-induced cerebral ischemia: histochemical and myeloperoxidase correlation with temporal changes in brain injury. J Cereb Blood Flow Metab. 2007;27(1):100-14.

3. Welniak-Kaminska M, Fiedorowicz M, Orzel J, Bogorodzki P, Modlinska K, Stryjek R, et al. Volumes of brain structures in captive wild-type and laboratory rats: 7T magnetic resonance in vivo automatic atlas-based study. PLoS One. 2019;14(4):e0215348.

4. Swanson RA, Morton MT, Tsao-Wu G, Savalos RA, Davidson C, Sharp FR. A semiautomated method for measuring brain infarct volume. J Cereb Blood Flow Metab. 1990;10(2):290-3.

---

## [Decision Letter · Decision Letter 1]

16 Jan 2024

PONE-D-23-12918R1The effects of fasting on acute ischemic infarcts in the ratPLOS ONE

Dear Dr. Schneider,

Thank you for submitting your manuscript to PLOS ONE. After careful consideration, we feel that it has merit but does not fully meet PLOS ONE’s publication criteria as it currently stands. Therefore, we invite you to submit a revised version of the manuscript that addresses the points raised during the review process.

Your manuscript, entitled "The effects of fasting on acute ischemic infarcts in the rat", has been reviewed. Your efforts to revise the manuscript are appreciated. However, the peer review process continues with minor changes. Please find the reviewer's commentary below.

We look forward to receiving your revised manuscript.

Kind regards,

Tanja Grubić Kezele, Ph.D., M.D.

Academic Editor

PLOS ONE

Journal Requirements:

Reviewers' comments:

Reviewer's Responses to Questions

**Comments to the Author**

1. If the authors have adequately addressed your comments raised in a previous round of review and you feel that this manuscript is now acceptable for publication, you may indicate that here to bypass the “Comments to the Author” section, enter your conflict of interest statement in the “Confidential to Editor” section, and submit your "Accept" recommendation.

Reviewer #2: All comments have been addressed

Reviewer #3: (No Response)

2. Is the manuscript technically sound, and do the data support the conclusions?

Reviewer #2: Yes

Reviewer #3: Partly

3. Has the statistical analysis been performed appropriately and rigorously? 

Reviewer #2: Yes

Reviewer #3: Yes

4. Have the authors made all data underlying the findings in their manuscript fully available?

Reviewer #2: Yes

Reviewer #3: No

5. Is the manuscript presented in an intelligible fashion and written in standard English?

Reviewer #2: Yes

Reviewer #3: Yes

6. Review Comments to the Author

Reviewer #2: The authors have addressed my comments. While there is potential for additional experiments to delve deeper into the mechanistic details of the findings, this reviewer believes that this study can be beneficial to the scientific community, particularly for further exploration of the effects of fasting on stroke outcomes.

Reviewer #3: The authors of this manuscript posit that fasting during the acute phase of cerebral infarction may lead to a reduction in the inflammatory response, thereby alleviating the impact of inflammation on cerebral infarction. To validate their hypothesis, the authors conducted rodent modeling trials. However, there are several aspects that require further consideration and elucidation by the authors.

1. What is the success rate of endothelin-1-induced focal cerebral ischemia modeling in rats in this experimental procedure? What were the specific criteria employed to ascertain the success of the modeling process? Were any of the samples that failed to achieve successful modeling included in the treatment group?

2. The rapamycin treatment group is mentioned in the Results section, yet no information regarding this group is provided in the Materials and Methods section of the manuscript.

7. PLOS authors have the option to publish the peer review history of their article (what does this mean?). If published, this will include your full peer review and any attached files.

Reviewer #2: No

Reviewer #3: No

---

## [Author Response · Author response to Decision Letter 1]

21 Jun 2024

Reviewer #2:

The authors have addressed my comments. While there is potential for additional experiments to delve deeper into the mechanistic details of the findings, this reviewer believes that this study can be beneficial to the scientific community, particularly for further exploration of the effects of fasting on stroke outcomes.

Reviewer #3:

The authors of this manuscript posit that fasting during the acute phase of cerebral infarction may lead to a reduction in the inflammatory response, thereby alleviating the impact of inflammation on cerebral infarction. To validate their hypothesis, the authors conducted rodent modeling trials. However, there are several aspects that require further consideration and elucidation by the authors.

1. What is the success rate of endothelin-1-induced focal cerebral ischemia modeling in rats in this experimental procedure? What were the specific criteria employed to ascertain the success of the modeling process? Were any of the samples that failed to achieve successful modeling included in the treatment group?

We thank the reviewers for their comments. The endothelin-1 stroke model is a well-validated and often-used method to induce ischemic stroke (1, 2). The stereotaxic microinjection of endothelin-1 induces a transient drop in blood flow lasting approximately 40 minutes and can be injected anywhere in the brain. This results in a focal lesion that can be used to study the effects of particular regions of the brain. We used cresyl violet staining for ischemia and immunohistochemical staining for blood-brain barrier integrity and neutrophil infiltration to directly and indirectly study focal lesions produced by endothelin-1. No animal had to be excluded in this study because of a potential failed lesion production. We amended the manuscript to make this clearer (lines 89-92).

2. The rapamycin treatment group is mentioned in the Results section, yet no information regarding this group is provided in the Materials and Methods section of the manuscript.

We are grateful to the reviewers for their valuable input. Although the original study involved comparing fasting, rapamycin treatment, and control groups, for this current publication, we have chosen to concentrate on our findings from the fasting group in comparison to the control group. The inclusion of rapamycin in the manuscript was an editing mistake, which has been rectified in the latest version.

References

1. Biernaskie J, Corbett D, Peeling J, Wells J, Lei H. A serial MR study of cerebral blood flow changes and lesion development following endothelin-1-induced ischemia in rats. Magn Reson Med. 2001;46(4):827-30.

2. Abeysinghe HCS, Roulston CL. A Complete Guide to Using the Endothelin-1 Model of Stroke in Conscious Rats for Acute and Long-Term Recovery Studies. Methods Mol Biol. 2018;1717:115-33.

---

## [Decision Letter · Decision Letter 2]

3 Jul 2024

The effects of fasting on acute ischemic infarcts in the rat

PONE-D-23-12918R2

Dear Dr. Schneider,

We’re pleased to inform you that your manuscript has been judged scientifically suitable for publication and will be formally accepted for publication once it meets all outstanding technical requirements.

Kind regards,

Tanja Grubić Kezele, Ph.D., M.D.

Academic Editor

PLOS ONE

Additional Editor Comments (optional):

Reviewers' comments:

Reviewer's Responses to Questions

**Comments to the Author**

1. If the authors have adequately addressed your comments raised in a previous round of review and you feel that this manuscript is now acceptable for publication, you may indicate that here to bypass the “Comments to the Author” section, enter your conflict of interest statement in the “Confidential to Editor” section, and submit your "Accept" recommendation.

Reviewer #3: All comments have been addressed

2. Is the manuscript technically sound, and do the data support the conclusions?

Reviewer #3: Yes

3. Has the statistical analysis been performed appropriately and rigorously? 

Reviewer #3: Yes

4. Have the authors made all data underlying the findings in their manuscript fully available?

Reviewer #3: Yes

5. Is the manuscript presented in an intelligible fashion and written in standard English?

Reviewer #3: Yes

6. Review Comments to the Author

Reviewer #3: All my previous concerns had been addressed. The authors have incorporated some of the statements they made in response to the reviewers into the manuscript itself. I recommend that the paper be published.

7. PLOS authors have the option to publish the peer review history of their article (what does this mean?). If published, this will include your full peer review and any attached files.

Reviewer #3: No

---

## [Editor Report · Acceptance letter]

29 Sep 2024

PONE-D-23-12918R2 

PLOS ONE

Dear Dr. Schneider, 

I'm pleased to inform you that your manuscript has been deemed suitable for publication in PLOS ONE. Congratulations! Your manuscript is now being handed over to our production team.

Kind regards, 

on behalf of

Prof. dr. Tanja Grubić Kezele 

Academic Editor

PLOS ONE